# Validating the Nepalese Short Attitudes Toward Mental Health Problems Scale (N-SATMHPS): A Culturally Sensitive Tool for Assessing Mental Health Stigma

**DOI:** 10.3390/ijerph22121884

**Published:** 2025-12-18

**Authors:** Dev Bandhu Poudel, Takashi Yoshioka, Rory Colman, Yasuhiro Kotera

**Affiliations:** 1Department of Humanities and Social Sciences, G. P. Koirala Memorial (Community) College, Kathmandu 44600, Nepal; 2Department of Humanities and Social Sciences, Brooklyn College, Kathmandu 44600, Nepal; 3Rupy’s International School (A-Levels)—Cambridge Associate School, Kathmandu 44600, Nepal; 4Health Technology Assessment Unit (Keio-HTA), Department of Preventive Medicine and Public Health, Keio University School of Medicine, Tokyo 160-8582, Japan; ty5733@gmail.com; 5Institute of Clinical Epidemiology (iCE), Showa University, Tokyo 142-8555, Japan; 6College of Health, Psychology and Social Care, University of Derby, Derby DE22 1GB, UK; r.colman@derby.ac.uk; 7Faculty of Medicine and Health Sciences, University of Nottingham, Nottingham NG7 2UH, UK; yasuhiro.kotera@nottingham.ac.uk; 8Center for Infectious Diseases Research and Education, The University of Osaka, Suita 565-0871, Japan; 9Department of Social Sciences, Azerbaijan University, Baku AZ1007, Azerbaijan

**Keywords:** attitudes toward mental illness, collectivist culture, discriminant validity, cultural validation, mental health literacy, mental health shame, mental health stigma, Nepal

## Abstract

**Highlights:**

**Public health relevance–How does this work relate to a public issue?**
Negative attitudes toward mental health hinder help-seeking in Nepal.Cultural factors such as shame and family honor influence public mental health perceptions.

**Public health significance–Why is this work of significance to public health?**
Provides a validated tool to measure mental health attitudes in Nepal.Identifies culturally specific dimensions of stigma relevant to collectivist societies.

**Public health implications–What are the key implications for practitioners, policy makers, and researchers?**
Can guide culturally tailored stigma-reduction programs and mental health interventions.Useful for monitoring public attitudes and evaluating mental health policies in Nepal.

**Abstract:**

(1) Background: Negative attitudes toward mental health problems remain a barrier for help-seeking, especially in collectivist, lower-middle-income countries like Nepal. While the Attitudes Towards Mental Health Problems Scale (ATMHPS) has been used globally, it has not been formally validated for Nepalese populations. This study aimed to culturally adapt and psychometrically validate a concise Nepalese version of the scale. (2) Methods: We conducted a cross-sectional study and recruited participants through an opportunity sampling method. We developed the Nepalese Short Version of the Attitudes Towards Mental Health Problems Scale (N-SATMHPS) using Dataset 1 (*n* = 384) and validated it with Dataset 2 (*n* = 803). Items were selected based on internal consistency indices. Fourteen items showing the strongest reliability were retained from the original seven subscales. A confirmatory factor analysis and internal consistency testing were used to assess psychometric properties. (3) Results: The N-SATMHPS showed a strong internal consistency (α = 0.74–0.92) and excellent model fit (χ^2^/df = 1.92, CFI = 0.982, TLI = 0.970, RMSEA = 0.049, SRMR = 0.026). Correlations with the full version ranged from r = 0.79 to 0.96. Discriminant validity with Mental Health Literacy Questionnaire—Young Adults (MHLQ-YA) showed weak but significant correlations, confirming construct distinction. (4) Conclusions: The scale captured key Nepalese cultural constructs, such as shame and family honor. It also aligned with collectivist cultural expectations. The N-SATMHPS demonstrates strong psychometric performance and cultural relevance. It is suitable for research and intervention work aimed at reducing stigma and improving mental health in Nepal.

## 1. Introduction

Negative attitudes toward mental health problems (ATMHP), often referred to as mental health stigma, remain a major barrier to seeking psychological support. This challenge is especially notable in low- and middle-income countries [1,2,3,4,5,6,7,8,9]. Negative attitudes are common even among caregivers and emerging medical professionals [10,11,12,13,14]. Despite growing awareness, mental illness continues to be perceived through a lens of shame, fear, and social exclusion, often rooted in cultural beliefs [15,16]. Such negative attitudes are compounded by societal norms that associate mental health issues with personal weakness or spiritual failings, leading to the discrimination and marginalization of affected individuals [17,18,19,20]. Therefore, measuring ATMHP is crucial, as it significantly contributes to stigma reduction and supports better mental health care. This is especially important in Nepal, a lower-middle-income country where mental health-related shame remains pervasive [2,5,6,7,8,21,22,23].

### 1.1. Measuring ATMHP

There are well-developed measurement scales to measure the attitudes of community members towards mental health issues. “Attitudes Towards Mental Health Problems (ATMHP)” developed by Gilbert et al. (2007) offers a comprehensive framework for assessing such attitudes, encompassing dimensions like internal and external shame and reflected shame [15]. In a Portuguese validation study of the ATMHP, the original model showed a poor fit in a confirmatory factor analysis (CFA), but an alternative model with an added factor demonstrated a good model fit. A further analysis confirmed that the revised version had good psychometric properties, supporting its suitability for assessing attitudes toward mental illness in Portuguese-speaking populations [24].

Later studies developed a 14-item short version of the ATMHPS (SATMHPS). It showed good internal consistency and replicated the original seven-factor structure, making it a reliable tool for assessing attitudes and shame toward mental health problems among UK university students [25]. Similarly, the Japanese short version of the Attitudes Towards Mental Health Problems Scale (J-SATMHPS) demonstrated a reliable seven-factor structure and cultural applicability. It highlighted differences between the UK and Japan in external and internal mental health-related shame [26]. The scale has already been used for measuring attitudes of college students in Nepal; however, the applicability of this scale in the Nepalese context has not been thoroughly examined [6].

### 1.2. Conceptual Framework of ATMHP Scale

The development of the ATMHP scale was grounded in a rich body of cross-cultural and psychosocial research examining how cultural norms shape perceptions of mental health, shame, and stigma. Culture acts as a framework through which individuals interpret their experiences, including attitudes toward mental illness [23,27,28]. Studies have emphasized obvious differences between Eastern and Western societies regarding the conceptualization of mental health, expectations of coping behaviors, and engagement with mental health services [29,30,31]. Research with British South Asian women underscored the role of shame—particularly tied to cultural values such as izzat (family honor)—in deterring help-seeking and promoting silence around psychological distress [32,33,34]. These findings informed the scale’s theoretical basis, incorporating dimensions of external shame (fear of social judgment), Internal Shame (self-criticism), and reflected shame (concern over bringing dishonor to one’s family) [35,36,37]. These constructs provide a culturally sensitive framework for assessing attitudes toward mental health.

### 1.3. Study Gap

While studies have utilized the ATMHP scale in various cultural settings without formal validation [6,38,39,40], formal validation has been performed in Portugal [24] and Japan [26]. Additionally, it has been used in cross-cultural studies, comparing Arabs, South Asians, and others living in the United Arab Emirates [41]; British Caucasians and Arabians [42]; and British Asians and non-Asians [15]. A shorter Japanese version of the ATMHPS (J-SATMHPS) has been developed and validated in a collectivistic culture [26]; however, there are no such validation studies in the South Asian sub-continent, including Nepal. Although Poudel et al. (2024) reported a high internal consistency (α = 0.94) and strong convergent validity (with Pearson correlations ranging from 0.66 to 0.86), the study was conducted without a proper validation of the tool, making the findings questionable [6]. Without such validation, the applicability and accuracy of the scale for capturing culturally embedded attitudes—such as those influenced by shame, family honor, and spiritual beliefs—remain uncertain. Though Nepal is a culturally collectivistic society similar to Japan, it also differs from the Japanese cultural context due to a complex interplay of cultural factors. Therefore, it is essential to develop and validate a shorter Nepalese version of the ATMHPS (N-SATMHPS).

### 1.4. Study Aim

This study aims to validate the ATMHP scale in the cross-cultural context of Nepal, ensuring its reliability and relevance for assessing attitudes toward mental health problems among Nepali populations. By doing so, it seeks to provide a culturally sensitive tool that can inform interventions aimed at reducing stigma and promoting mental well-being in Nepal.

This study can contribute to the Nepalese mental health field by validating the ATMHP scale for use in Nepal. It ensures the tool’s cultural relevance and reliability in assessing negative attitudes rooted in societal shame, stigma, and family honor, which are prominent in Nepalese culture. By offering a validated, context-sensitive instrument, this study lays the groundwork for more effective stigma reduction interventions and evidence-based mental health strategies tailored to the Nepalese population.

## 2. Materials and Methods

### 2.1. Research Design

Two different cross-sectional survey designs were utilized simultaneously to develop the Nepalese Short Version of the Attitudes Towards Mental Health Problems Scale (N-SATMHPS) Appendix A; to evaluate the structural validity of the N-SATMHPS; and to assess the discriminant validity between the N-SATMHPS and Mental Health Literacy Questionnaire—Young Adult (MHLQ–YA; Dias et al., 2018) [43], including their respective dimensions. Dataset 1, from a previous study by Poudel et al. (2024) [6], was utilized to develop the scale, while Dataset 2 was used to evaluate the structure in a Confirmatory Factor Analysis (CFA).

### 2.2. Participants and Procedures

For Dataset 1, the required sample size was determined using Cochran’s formula for a population with an unknown proportion: n0 = (Z2pq)/e2, where n0 is the sample size, Z is the z-score for the desired confidence level, p is the estimated proportion of the population, q is 1 − p, and e is the desired level of precision (i.e., marginal error) [44]. Assuming a 95% confidence level (Z = 1.96), a proportion estimate of 0.5 (p = 0.5, q = 0.5), and a 5% margin of error (e = 0.05), the minimum required sample size was calculated as ((1.96)2 (0.5) (0.5))/(0.05)2 = 385 [44]. Based on this, a total of 384 participants aged 18 to 24 years were included (1 was excluded due to sensitivity) from the Chitwan and Kathmandu districts (i.e., Dataset 1). This sample, also used in a previous study [6], was used to develop the tool in the current study. For Dataset 2, we included 803 participants from across Nepal, encompassing both students and individuals from the general population from very remote areas such as Karnali in the west and Jhapa in the east, including Kathmandu and Chitwan, which were used for CFA. The sample comprised diverse demographic characteristics, including various demographic components such as gender and religious and ethnic groups within the country.

Opportunity sampling, a non-probability method suitable for accessing readily available populations, was employed. College students were included because they are a key group for understanding and shaping societal attitudes toward mental health [45,46], and general individuals were included for broader representation.

### 2.3. Instruments

Two standardized self-report instruments were employed via online Google forms and paper-based questionnaires: the ATMHP scale [15] and MHLQ-YA [43]. The tools were translated into Nepali following rigorous cross-cultural adaptation procedures to minimize language-related bias. The forward translation was performed by a professional bilingual English language teacher, and the back translation was conducted by another bilingual language English teacher with translation experience. The translated version was then evaluated by five additional English language teachers to identify linguistic ambiguities. A psychology lecturer with expertise in mental health reviewed the translation for accuracy and cultural relevance, and a Nepali language expert checked the spelling, punctuation, and grammar. These steps ensured both linguistic and conceptual equivalence in the adapted tools.

Google forms were distributed via platforms such as Facebook, LinkedIn, and Instagram. For the paper-based method, participants were approached directly on campuses. The online survey was administered between 26 June and 24 October 2021, while the paper–pencil survey was conducted from 27 September to 4 October 2021.

### 2.4. Ethical Consideration

Ethical approval for the first study (Dataset 1, *n* = 384) was obtained from the Nepal Health Research Council under Ethics Review Board protocol registration no. 309/2021 (ref. no. 3543, approval date: 15 June 2021) [6], and for the second study (Dataset 2, *n* = 803), we obtained approval from college authorities according to their rules and regulations; college–1 (reference no. 151/2077/78; date: 20 June 2021), college–2 (reference no. 105/077/078; date: 22 June 2021); college–3 (reference no. 423; date: 1 October 2021); college–4 (reference no. 433; 4 October 2021), and college–5 (Dispatch no. 544; 4 October 2021). Informed consent was obtained after participants were briefed on confidentiality, data security, and their right to withdraw at any time. This study adhered to the ethical principles of the Declaration of Helsinki, ensuring participant rights and autonomy throughout [47]. Licensed and free software tools (MS Office—16, Stata version 18, semdiag and Google Forms/Sheets) were used for data collection, analysis, the path diagram creation, and manuscript preparation [48]. Data supporting the findings of this study are available from the corresponding author upon reasonable request.

### 2.5. Data Analysis

Data were analyzed using Stata 18.0, following the methods of a similar UK study [25], and the path diagram was created using semdiag [48]. Dataset 1 (*n* = 384) was used for item reduction, and Dataset 2 (*n* = 803) was used for validation. Outliers in Dataset 1 were removed by visually inspecting Total Scores; perfect scores (105) were excluded to improve sensitivity. Descriptive statistics summarized demographic data. Pearson correlations identified two items with the highest inter-item correlations within each of the seven ATMHPS subscales, producing a 14-item N-SATMHPS. Internal consistency was assessed using Cronbach’s alpha, and item–total correlations evaluated the item performance. Convergent validity was tested by correlating subscale totals between the original and shortened scales. Confirmatory factor analysis (CFA) was conducted to assess model fit using χ^2^/df, CFI, TLI, RMSEA, and SRMR as fit indices. Dataset 2 was used to replicate the procedures and confirm the reliability and structural validity of the N-SATMHPS. Based on established guidelines for structural equation modeling, we considered a model acceptable if the comparative fit index (CFI) and Tucker–Lewis index (TLI) were ≥0.90, and if the root mean square error of approximation (RMSEA) and standardized root mean square residual (SRMR) were ≤0.08 [49].

The 14 items were selected from the initial item pool based on their high inter-item correlations, ensuring that retained items were strongly related to each other and represented the core construct. In addition to correlation values, the theoretical relevance and coverage of the construct dimensions guided the selection. Exploratory factor analysis or item response theory methods were not applied at this stage, as the aim was a preliminary selection of items based on both statistical strength and conceptual considerations.

## 3. Results

### 3.1. Dataset Overview

We developed the N-SATMHPS using Dataset 1 (*n* = 384) and validated it with Dataset 2 (*n* = 803). Each dataset contained the full ATMHPS items, labeled atmhp1 to atmhp35. One response in Dataset 1 with a perfect Total Score (105 points) was excluded, as it may reflect an extreme response style such as acquiescence bias or social desirability rather than genuine attitudes [50]. This exclusion followed best practices in psychometric research to improve the sensitivity and interpretability of scale performance. The analytical methods were based on prior work by Kotera et al. (2023) [25], and analyses were conducted using Stata 18.0. A path diagram was generated using the semdiag package [48].

### 3.2. Confirmation of Background Factors

Next, we examined the background factors of the participants in each dataset. Demographic characteristics differed between the two datasets. Participants in Dataset 1 were younger, predominantly female, and mostly students. In contrast, Dataset 2 included a broader age range with more occupational diversity (Table 1 and Table 2).

### 3.3. Extraction of Two Questions for Correlation Analysis for N-SATMHPS

A correlation analysis was conducted on the group in Dataset 1 (Table 3). We evaluated the correlations between the questions within each subscale and selected the two items with the highest correlation.

### 3.4. The Performance Evaluation of the Developed N-SATMHPS

Next, we evaluated the correlation (item–total correlations) and *p*-values of the selected questions and each subscale. The results are presented in Table 4.

Unlike the benchmark validation study [25], the correlation coefficients ranged from 0.59 to 0.83, showing low values for some items. Next, we calculated the average, standard deviation, and α for each item (scored 0–3) for each subscale (Table 5).

The alpha value (α) had a reliability of 0.79–0.92 in the ATMHPS and 0.71–0.89 in the N-SATMHPS, which was not as reliable as the reference paper. The table below shows the correlation of the sums of the subscales for the ATMHPS and N-SATMHPS (Table 6).

Except for the Family-Reflected Shame, a strong correlation was obtained.

Finally, a path diagram was created, and the goodness of fit of the model was evaluated (Figure A1 in Appendix B).

The confirmatory factor analysis (CFA) indicated a good model fit, with all fit indices falling within acceptable ranges. The chi-square to degrees of freedom ratio (χ^2^/df) was 107.32/56 = 1.92, suggesting an adequate fit. The comparative fit index (CFI) was 0.98, and the Tucker–Lewis index (TLI) was 0.97, both exceeding the recommended threshold of 0.90. Additionally, the root mean square error of approximation (RMSEA) was 0.049, and the standardized root mean square residual (SRMR) was 0.026, both indicating a good fit according to conventional guidelines.

### 3.5. Validation

We developed and validated the N-SATMHPS using Dataset 1 and Dataset 2, respectively. First, we confirmed the internal consistency. The results are presented in Table 7.

The results showed that the alpha value for the ATMHPS was 0.84–0.94, and for the N-SATMHPS it was 0.74–0.92, indicating that the coefficients for the shortened versions were generally lower than those of the current versions. However, the alpha values themselves were within an acceptable range.

Finally, as in the development stage, the following table confirms the correlation of the sums of each subscale of the N-SATMHPS (Table 8).

The results showed that the correlation coefficients were generally higher than those from the development stage.

### 3.6. Discriminant Validity

Several significant but weak correlations were observed between the MHLQ-YA factors and N-SATMHPS subscales, supporting discriminant validity. Notably, MHLQ-YA’s Factor 1 was positively correlated with the N-SATMHPS’s Community Attitudes and Total Score. Factor 2 showed negative correlations with Family Attitudes, Family External Shame, Self-Reflected Shame, and the Total Score. Factor 3 was negatively associated with Family Attitudes, Family External Shame, Internal Shame, and the Total Score. No significant associations emerged for Factor 4 or the MHLQ-YA Total Score. The details are presented in Table 9.

## 4. Discussion

The development and validation of the N-SATMHPS represent a significant advancement in culturally sensitive mental health assessment tools. We aimed to create a concise yet reliable instrument to measure attitudes towards mental health problems within the Nepali context, ensuring both psychometric robustness and practical applicability.

### 4.1. Model Fit

The N-SATMHPS demonstrated satisfactory internal consistency across subscales. While slightly lower than the reliability scores of the full ATMHPS and the short-form J-SATMHPS, the values remain within acceptable thresholds for psychological tools, supporting the scale’s reliability [15,26]. The CFA supported the seven-factor structure of the N-SATMHPS, with fit indices indicating a good model fit. These findings are consistent with the CFA results for the SATMHPS and J-SATMHPS, both of which demonstrated strong model fits [25,26].

### 4.2. Correlation Analyses

The N-SATMHPS subscales exhibited strong correlations with their corresponding subscales in the full ATMHPS, ranging from 0.79 to 0.96. This consistency underscores the N-SATMHPS’s capability to effectively capture the constructs measured by the full scale [15].

The discriminant validity of the N-SATMHPS was assessed through correlations with the MHLQ-YA [43]. Significant but weak correlations were observed between certain N-SATMHPS subscales and MHLQ-YA factors, suggesting that while related, the constructs measured by the two instruments are distinct. For instance, MHLQ-YA’s Factor 1 (Knowledge of Mental Health Problems) showed a positive correlation with the Community Attitudes subscale of the N-SATMHPS, suggesting that greater knowledge of mental health increases negative Community Attitudes, whereas Factor 2 (Erroneous Beliefs/Stereotypes) exhibited negative correlations with several N-SATMHPS subscales, indicating that higher awareness of Erroneous Beliefs/Stereotypes promotes positive attitudes. Studies have found that mental health literacy reduces mental health shame [51,52,53]. However, Poudel et al. (2024) have found no relation between mental health knowledge and mental health shame [6].

### 4.3. Cultural Considerations and Implications

The adaptation of the ATMHPS into the Nepali context required careful consideration of the cultural nuances related to mental health stigma and shame. The strong internal consistency and model fit indices suggest that the N-SATMHPS effectively captures culturally relevant attitudes towards mental health problems in Nepal.

The successful cultural adaptation of the ATMHPS into the Nepalese context underscores the scale’s utility in collectivist societies, where community and family perceptions deeply influence individual behavior [54]. Unlike Western cultures, which generally prioritize individual autonomy [55], Nepalese society often views mental health through a communal lens, where family honor (izzat) and societal expectations play a critical role [32]. Concepts such as reflected shame (i.e., feeling disgrace on behalf of the family) and external shame (i.e., fear of judgment by the community) are embedded in everyday attitudes. In our study, these constructs remained relevant and strongly endorsed, validating the theoretical structure of the N-SATMHPS. Moreover, despite some subcultural diversity in Nepal, our results showed consistency across gender, caste, and religious affiliations, suggesting that the scale captures common cultural attitudes rather than group-specific perspectives. The validation of the N-SATMHPS contributes not only to Nepalese mental health research but also offers a model for culturally contextualizing global tools in diverse settings.

Assessing mental health shame is particularly valuable because it can directly in-form interventions that encourage help-seeking and self-compassion [56]. High levels of shame often prevent individuals from accessing mental health services due to fear of judgment or loss of social standing [57]. By identifying these attitudes, targeted psychoeducation and compassion-focused approaches can be developed to reduce shame, normalize help-seeking, and promote a more accepting self-view. Evidence suggests that less shame and greater self-compassion are linked to a greater willingness to seek professional support and improved mental health outcomes [58]. Therefore, the N-SATMHPS not only measures attitudes but also offers practical insights for designing culturally appropriate mental health interventions in Nepal.

### 4.4. Limitations

This study has several limitations. First, both datasets were obtained through opportunity sampling, with a predominance of college students. This may limit the generalizability of the findings to the wider Nepalese population, particularly rural residents and non-educated groups, who may hold different attitudes toward mental health [59]. Second, although the translation and cross-cultural adaptation process ensured linguistic and conceptual equivalence, this study did not collect qualitative feedback from participants or experts to examine how constructs such as reflected shame or izzat are locally interpreted. Third, this study did not assess the test–retest reliability or convergent validity with other stigma or shame measures, limiting insights into the scale’s stability and its relationship with related constructs [60,61]. Fourth, the reliance on self-report measures introduces potential response biases, especially given the sensitive nature of mental health stigma [26]. Fifth, no qualitative methods were used, which could have deepened the understanding of how shame and stigma are experienced in context. Finally, the clinical and applied utility of the N-SATMHPS was not evaluated. Future studies should address these remaining aspects to further establish the scale’s psychometric robustness and utility.

## 5. Conclusions

The N-SATMHPS emerges as a reliable and valid instrument for assessing attitudes towards mental health problems in Nepal [61]. Its brevity and cultural relevance make it a practical tool for both research and clinical settings. By facilitating the assessment of mental health attitudes, the N-SATMHPS can contribute to the development of targeted interventions aimed at reducing stigma and promoting mental well-being within the Nepali context.

## Figures and Tables

**Table 1 ijerph-22-01884-t001:** Background characteristics of participants (Dataset 1, *n* = 384).

Variables	n (%)
Age (Mean, SD)	20.7 (1.7)
Gender (Female)	236 (61.6)
Caste/Ethnicity (Brahmin/Kshetri)	269 (70.6)
Religion (Hindu)	344 (89.6)
Districts (Chitwan)	348 (90.4)
Marital Status (Unmarried)	369 (95.8)
Academic Qualification (Bachelor’s Degree)	325 (84.4)

**Table 2 ijerph-22-01884-t002:** Background characteristics of participants (Dataset 2, *n* = 803).

Variables	n (%)	Missing (n)
Age (Mean, SD)	27.9 (8.7)	21
Gender (Female)	351 (44.4)	12
Caste/Ethnicity (Brahmin/Kshetri)	524 (66.0)	9
Religion (Hindu)	660 (84.0)	17
Districts (Chitwan)	247 (32.1)	34
Marital Status (Unmarried)	489 (61.6)	9
Academic Qualification (Bachelor’s Degree)	352 (44.7)	15
Working Status (Student)	287 (36.0)	6

**Table 3 ijerph-22-01884-t003:** Correlation between items in each factor of the ATMHPS for Dataset 1.

Community Attitudes							
	Item01	Item02	Item03	Item04			
Item01	—						
Item02	0.52	—					
Item03	0.47	0.55	—				
Item04	0.45	0.37	0.54	—			
Family Attitudes							
	Item05	Item06	Item07	Item08			
Item05	—						
Item06	0.57	—					
Item07	0.51	0.57	—				
Item08	0.48	0.55	0.71	—			
Community External Shame							
	Item09	Item10	Item11	Item12	Item13		
Item09	—						
Item10	0.71	—					
Item11	0.65	0.75	—				
Item12	0.61	0.70	0.80	—			
Item13	0.58	0.65	0.67	0.74	—		
Family External Shame							
	Item14	Item15	Item16	Item17	Item18		
Item14	—						
Item15	0.73	—					
Item16	0.69	0.80	—				
Item17	0.55	0.59	0.64	—			
Item18	0.63	0.66	0.71	0.70	—		
Internal Shame							
	Item19	Item20	Item21	Item22	Item23		
Item19	—						
Item20	0.77	—					
Item21	0.46	0.45	—				
Item22	0.60	0.65	0.45	—			
Item23	0.54	0.56	0.39	0.65	—		
Family-Reflected Shame							
	Item24	Item25	Item26	Item27	Item28	Item29	Item30
Item24	—						
Item25	0.71	—					
Item26	0.58	0.57	—				
Item27	0.59	0.60	0.57	—			
Item28	0.33	0.33	0.40	0.42	—		
Item29	0.37	0.39	0.38	0.46	0.70	—	
Item30	0.36	0.37	0.40	0.51	0.54	0.69	—
Self-Reflected Shame							
	Item31	Item32	Item33	Item34	Item35		
Item31	—						
Item32	0.66	—					
Item33	0.67	0.70	—				
Item34	0.68	0.67	0.72	—			
Item35	0.61	0.58	0.59	0.67	—		

**Table 4 ijerph-22-01884-t004:** Correlation coefficients and *p*-values of the N-SATMHPS.

N-SATMHPS Items	Subscale	Correlation Coefficients	*p*
2	Community Attitudes	0.59	<0.001
3	Community Attitudes	0.65	<0.001
7	Family Attitudes	0.71	<0.001
8	Family Attitudes	0.69	<0.001
11	Community External Shame	0.83	<0.001
12	Community External Shame	0.82	<0.001
15	Family External Shame	0.80	<0.001
16	Family External Shame	0.82	<0.001
19	Internal Shame	0.73	<0.001
20	Internal Shame	0.76	<0.001
24	Family-Reflected Shame	0.63	<0.001
25	Family-Reflected Shame	0.64	<0.001
33	Self-Reflected Shame	0.78	<0.001
34	Self-Reflected Shame	0.80	<0.001

**Table 5 ijerph-22-01884-t005:** Comparison of mean, SD, and correlation between full ATMHP and N-SATMHPS.

	Mean		SD		α	
	ATMHPS	N-SATMHPS	ATMHPS	N-SATMHPS	ATMHPS	N-SATMHPS
Community Attitudes	1.60	1.64	0.81	0.88	0.79	0.71
Family Attitudes	0.65	0.54	0.73	0.79	0.84	0.83
Community External Shame	1.21	1.26	0.92	1.03	0.92	0.89
Family External Shame	0.45	0.39	0.67	0.70	0.91	0.89
Internal Shame	0.73	0.60	0.74	0.83	0.86	0.87
Family-Reflected Shame	1.11	0.71	0.76	0.82	0.87	0.83
Self-Reflected Shame	1.11	1.11	0.92	1.00	0.90	0.84

**Table 6 ijerph-22-01884-t006:** Correlation between full ATMHP and N-SATMHPS.

	r
Community Attitudes	0.90
Family Attitudes	0.90
Community External Shame	0.94
Family External Shame	0.93
Internal Shame	0.89
Family-Reflected Shame	0.79
Self-Reflected Shame	0.94

**Table 7 ijerph-22-01884-t007:** Comparison of reliability scores (α) between full ATMHP and N-SATMHPS.

Factors	α	
	ATMHPS	N-SATMHPS
Community Attitudes	0.84	0.74
Family Attitudes	0.87	0.84
Community External Shame	0.94	0.91
Family External Shame	0.94	0.92
Internal Shame	0.89	0.89
Family-Reflected Shame	0.91	0.91
Self-Reflected Shame	0.93	0.89

**Table 8 ijerph-22-01884-t008:** Correlations of factors in N-SATMHPS.

	r
Community Attitudes	0.94
Family Attitudes	0.91
Community External Shame	0.96
Family External Shame	0.96
Internal Shame	0.90
Family-Reflected Shame	0.85
Self-Reflected Shame	0.95

**Table 9 ijerph-22-01884-t009:** Discriminant validity between N-SATMHPS and MHLQ-YA.

	MHLQ-YA Factor 1 Knowledge of Mental Health Problems	MHLQ-YA Factor 2 Erroneous Beliefs/Stereotypes	MHLQ-YA Factor 3 First Aid Skills and Help-Seeking Behavior	MHLQ-YA Factor 4 Self-Help Strategies	MHLQ-YA Total Score
	Correlation Coefficients	Correlation Coefficients	Correlation Coefficients	Correlation Coefficients	Correlation Coefficients
N-SATMHPS Community Attitudes	0.19 **	−0.07	−0.03	−0.06	0.04
N-SATMHPS Family Attitudes	0.07	−0.13 *	−0.14 **	−0.05	−0.08
N-SATMHPS Community External Shame	0.12 *	0.01	−0.06	0.01	0.04
N-SATMHPS Family External Shame	0.03	−0.11 *	−0.12 *	0.00	−0.07
N-SATMHPS Internal Shame	0.05	−0.04	−0.13 **	0.03	−0.03
N-SATMHPS Family-Reflected Shame	0.06	−0.04	−0.03	−0.01	0.01
N-SATMHPS Self-Reflected Shame	−0.03	−0.13 *	−0.09	0.04	−0.09
N-SATMPHS Total Score	0.11 *	−0.10 *	−0.13 *	−0.00	−0.04

* *p* < 0.05, ** *p* < 0.01

## Data Availability

Certain data are available upon reasonable request from the corresponding author.

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
