# Peer review of "Validating the Nepalese Short Attitudes Toward Mental Health Problems Scale (N-SATMHPS): A Culturally Sensitive Tool for Assessing Mental Health Stigma"

_ijerph, 2025, doi:10.3390/ijerph22121884_

Round 1

Reviewer 1 Report

Comments and Suggestions for Authors

I believe that the abstract has a clear and coherent structure in accordance with the IMRaD format, adequately describing the context, the psychometric methods used, and the main results, which demonstrate the solid validity and reliability of the adapted instrument. However, I believe there is room for improvement in the writing, such as the presence of long sentences and line breaks that affect fluidity, the excessive use of acronyms, and certain imprecise expressions. I also believe it would be useful to better specify the item selection criteria and provide a clearer interpretation of the results.

As for the introduction, it adequately fulfils its function of contextualising the problem under study, clearly presenting the relevance of negative attitudes towards mental health (NATMH) as a barrier to seeking help, especially in low-to-middle-income contexts such as Nepal. I find the text to be well structured and substantiated, as it describes the theoretical and empirical background of the ATMHP scale, its international adaptations, and the conceptual framework based on the cultural influence of shame and family honour. I note, however, that the section could be improved with more concise wording, avoiding repetitions and long sentences. I also recommend strengthening the connection between the subsections and highlighting more clearly the research gap that justifies the study.

Regarding the Materials and Methods section, it presents a detailed and coherent description of the design, participants, instruments, and statistical procedures used, reflecting adequate methodological rigour. I believe that the use of two independent databases (Dataset 1 and Dataset 2) strengthens the validity of the study, as it allows for both the development and validation of the instrument through confirmatory factor analysis (CFA). However, I note that there is an excess of technical information that could be summarised to improve clarity. The description of the translation and cultural adaptation process is comprehensive and demonstrates a strong focus on linguistic and conceptual equivalence, although I believe it could be summarised without losing methodological detail. I also recommend clarifying the type of sampling (referred to as ‘opportunity sampling’) and justifying its choice.

I consider that the Results section presents a comprehensive and well-structured development of the validation process of the abbreviated Nepalese version of the ATMHPS scale (N-SATMHPS), demonstrating solid psychometric rigour. I note that the authors clearly describe the composition and demographic differences between the two data sets used (n=384 and n=805), as well as the item selection procedures, internal correlations, and confirmatory factor analysis (CFA).

The discussion presents a solid and well-structured analysis of the validity, reliability, and cultural relevance of the N-SATMHPS, highlighting its contribution to the assessment of attitudes toward mental health in non-Western contexts. I appreciate that the psychometric results are interpreted consistently, noting that the scale maintains adequate internal consistency and good factorial fit, comparable to previous versions such as the SATMHPS and the J-SATMHPS. I note and appreciate that the importance of cultural adaptation is correctly highlighted, explaining how sociocultural factors specific to Nepal (such as the centrality of the family, honour and reflected or external shame) influence attitudes towards mental health problems, which reinforces the relevance of the instrument.

With regard to the bibliographical references, I have noted several self-citations, and after analysing the self-citations included in the manuscript, I consider that most of them are relevant and necessary, as they refer to previous work by the same research group related to the development, validation and cross-cultural adaptation of the scale used. These references provide adequate methodological and conceptual support, strengthening the justification for the instrument used and the coherence of the theoretical framework. However, some self-citations—particularly references 56, 57, and 59—are more tangentially related to the study's objectives and could be removed without affecting the theoretical soundness or central argument of the article. We therefore suggest a selective review of these references in order to improve the accuracy and balance of the bibliography.

In my opinion, the manuscript addresses a relevant and current topic in the field of mental health, demonstrating adequate methodological rigour and making a clear contribution to knowledge about attitudes towards mental health problems in cross-cultural contexts. However, it is recommended that the aspects highlighted throughout the report be reviewed, especially those related to writing, conceptual accuracy, and reference selection, in order to strengthen the clarity, coherence, and theoretical soundness of the work. Overall, this is a study with the potential to be published after minor revision and careful improvement of formal and methodological details.

Reviewer 2 Report

Comments and Suggestions for Authors

Thank you for the opportunity to review this clearly presented and impactful study. 

General comments

This is a clear, concise and useful study. Please see the uploaded file with comments on minor errors in the English expression, spelling and punctuation.

While validation of psychometric scales is not particularly novel research, this scale is important to the Nepalese population for detecting and addressing stigma and barriers to service-seeking behaviours.

Introduction

The introduction is extremely clear and concise, showing the pedigree of this type of assessment in collectivist cultures, and its specific relevance to the Nepalese context. 

Methods

The methods were clear, logical and comprehensive. Details justifying sample selection (and exclusions), translation processes and statistical measures were clearly stated and easy to follow.  This study could be easily replicated.

Discussion

The discussion highlights the likely sub-cultural generalisability in Nepal, whilst limitations included validation with young adults, lack of test-retest validation and real-world utility.  The utility of this tool to start to address reflected and external shame seems important and useful.  I look forward to reading further studies of its application and exploration of qualitative dimensions.

Reviewer 3 Report

Comments and Suggestions for Authors

This manuscript reports the development and psychometric validation of a short version of the ATMHPS for use in Nepal N-SATMHPS. The study addresses a relevant and timely topic in global mental health, namely the cultural adaptation of stigma measures for collectivist and low-middle-income settings. The work is generally well structured, clearly written, and grounded in solid theoretical and cross-cultural frameworks. The results suggest that the N-SATMHPS demonstrates good reliability, strong factorial validity, and adequate discriminant validity, while successfully capturing culture-specific aspects such as shame and family honor.

Overall, the manuscript makes a valuable contribution to the field of cross-cultural psychometrics and stigma research. However, some methodological clarifications and editorial refinements are needed before publication.

Both datasets were obtained through opportunity sampling, largely composed of college students. This limits the representativeness of the findings. The authors should discuss more explicitly how this sampling approach may affect generalizability to the wider Nepalese population, especially rural and non-educated groups.

Given the demographic differences between Dataset 1 and Dataset 2 (age, gender, occupation), it would strengthen the paper to include at least a brief sensitivity or multi-group CFA to verify measurement invariance across key subgroups (e.g., gender, age, education).

Item reduction process: The procedure for selecting the 14 items seems based solely on inter-item correlations. It would be helpful to clarify the precise criteria (e.g., threshold values, theoretical considerations) and to justify why exploratory factor analysis or item-response theory methods were not applied. A short paragraph in the Methods section could resolve this.

While internal consistency and discriminant validity were well addressed, test–retest reliability and convergent validity (with other stigma or shame measures) were not assessed. The authors should acknowledge this more explicitly as a limitation and outline plans for future validation studies.

The translation process is carefully described, but the discussion of cultural adaptation would benefit from deeper reflection. For instance, qualitative feedback from participants or experts could have provided insight into how constructs like reflected shame or izzat were interpreted locally. The authors might wish to note this as a potential future step.

Some tables are duplicated or contain minor inconsistencies (e.g., “Family External Shame” appears twice in Table 7). Please review the numbering and ensure all figures are properly embedded rather than listed as placeholders (“Please insert Figure 1 here”).

The manuscript is generally well written, though minor grammatical and formatting issues remain (e.g., occasional repetitions, missing commas). A careful proofreading would be advisable before final submission.

The ethics section is clear and comprehensive. Still, for transparency, it may be useful to specify whether any data sharing restrictions apply and under what conditions the dataset can be accessed.
